# SF-V: Single Forward Video Generation Model

**Zhixing Zhang**[1,2*]    **Yanyu Li**[1]    **Yushu Wu**[1]    **Yanwu Xu**[1]    **Anil Kag**[1]
**Ivan Skorokhodov**[1]    **Willi Menapace**[1]    **Aliaksandr Siarohin**[1]    **Junli Cao**[1]
**Dimitris Metaxas**[2]    **Sergey Tulyakov**[1]    **Jian Ren**[1†]
[1]Snap Inc.    [2] Rutgers University

Project Page: https://snap-research.github.io/SF-V

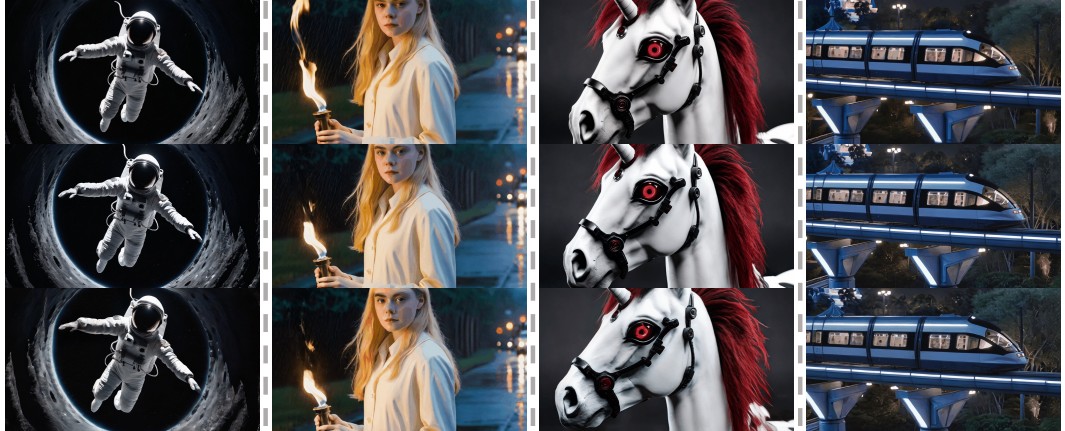

Figure 1: Example generation results from our *single*-step image-to-video model. Our model can generate high-quality and motion consistent videos by only performing the sampling *once* during inference. Please refer to our webpage for whole video sequences.

## Abstract

Diffusion-based video generation models have demonstrated remarkable success in obtaining high-fidelity videos through the iterative denoising process. However, these models require multiple denoising steps during sampling, resulting in high computational costs. In this work, we propose a novel approach to obtain *single*-step video generation models by leveraging adversarial training to fine-tune pre-trained video diffusion models. We show that, through the adversarial training, the multi-steps video diffusion model, *i.e.*, Stable Video Diffusion (SVD), can be trained to perform *single* forward pass to synthesize high-quality videos, capturing both temporal and spatial dependencies in the video data. Extensive experiments demonstrate that our method achieves competitive generation quality of synthesized videos with significantly reduced computational overhead for the denoising process (*i.e.*, around $23\times$ speedup compared with SVD and $6\times$ speedup compared with existing works, with even better generation quality), paving the way for real-time video synthesis and editing.

---

*Work done during an internship at Snap Inc.

†Corresponding author.

38th Conference on Neural Information Processing Systems (NeurIPS 2024).

# 1 Introduction

Video generation is experiencing unprecedented advancements by leveraging large-scale denoising diffusion probabilistic models [1, 2] to create photo-realistic frames with natural and consistent motion [3, 4], revolutionizing various fields, such as entertainment and digital content creation [5, 6].

Early efforts on image generation show that diffusion models have the significant capabilities when scaled-up to generate diverse and high-fidelity content [1, 2]. Additionally, these models benefit from a stable training and convergence process, demonstrating a considerable improvement over their predecessors, *i.e.*, generative adversarial networks (GANs) [7]. Therefore, many studies on video generation are built upon the diffusion models. Some of them utilize the pre-trained image diffusion models for video synthesis through introducing temporal layers to generate high-quality video clips [8, 9, 10, 11]. Inspired by this design paradigm, numerous video generation applications have emerged, such as animating a given image with optional motion priors [12, 13, 14, 15], generating videos from natural language descriptions [16, 17, 5], and even synthesizing cinematic and minutes-long temporal-consistent videos [18, 4].

Despite the impressive generative performance, video diffusion models suffer from tremendous computational costs, hindering their widespread and efficient deployment. The iterative nature of the sampling process makes video diffusion models significantly slower than other generative models (*e.g.*, GANs [19, 20]). For instance, in our benchmark, it only takes $0.3$ seconds to perform a single denoising step using the UNet from the Stable Video Diffusion (SVD) [13] model to generate $14$ frames on one NVIDIA A100 GPU, while consuming $10.79$ seconds to run the UNet with the conventional 25-step sampling.

The significant overhead introduced by iterative sampling highlights the necessity to generate videos in fewer steps while maintaining the quality of multi-step sampling. Recent works [21, 22, 23] extend consistency training [24] to video diffusion models, offering two main benefits: reduced total runtime by performing fewer sampling steps and the preservation of the pre-trained ordinary differential equation (ODE) trajectory, allowing high-quality video generation with fewer sampling steps (*e.g.*, $8$ steps). Nevertheless, these approaches still struggle to achieve *single*-step high-quality video generation.

On the other hand, distilling image diffusion models into one step via adversarial training have shown promising progress [25, 26, 27, 28, 29]. However, scaling up such approaches for video diffusion model training to achieve single-step generation has not been well studied. In this work, we leverage adversarial training to obtain an image-to-vide o generation model that requires only *single*-step generation, with the contributions summarized as follows:

- We build the framework to fine-tune the pre-trained state-of-the-art video diffusion model (*i.e.*, SVD) to be able to generate videos in *single* forward pass, greatly reducing the runtime burden of video diffusion model. The training is conducted through adversarial training on the latent space.

- To improve the generation quality (*e.g.*, higher image quality and more consistent motion), we introduce the discriminator with spatial-temporal heads, preventing the generated videos from collapsing to the conditional image.

- We are the first to achieve one-step generation for video diffusion models. Our one-step model demonstrates superiority in FVD [30] and visual quality. Specifically, for the denoising process, our model achieves around $23\times$ speedup compared with SVD and $6\times$ speedup compared with exiting works, with even better generation quality.

# 2 Related Work

**Video Generation** has been a long studied problem, aiming for high-quality image generation and consistent motion synthesis. Early efforts in this domain utilize adversarial training [31, 32]. Though extensively investigated, the trained models still suffer from low resolution, limited generated sequences, and inconsistent motion. Recent studies leverage denoising diffusion probabilistic models [1, 33, 34] to scale the video generators up to billions of model parameters, achieving high-fidelity generation sequences [35, 36, 37, 38, 39, 5, 4, 3, 18]. Nonetheless, the tremendous computation cost of video diffusion models hinders their wide deployment. It takes tens of seconds to generate a

single video batch even for high-tier server GPUs. Consequently, the reduction of denoising steps [21, 40, 22] is pivotal to efficient video generation, which linearly scales down the total runtime.

**Step Distillation of Diffusion Models.** Initially developed upon image diffusion models, progressive distillation [41, 42] aims to distill a less-step student mimicking the full-step counterpart. Specifically, at each step, the student learns to predict a teacher location in the ODE flow, resulting in fewer required denoising steps during inference time. Latent Consistency Models (LCM) [24, 43, 44, 45, 46, 47, 48] instead proposes to refine the prediction objective into clean data, and achieves high-fidelity generation with fewer ($2 \sim 4$) steps. Rectified flow [49, 50] progressively straights the ODE flow where each denoising step becomes a substitution of a long trajectory. UFOGen [25], ADD [27], and its latent-space successor LADD [28] further incorporate adversarial loss to distill teacher signal into the few-step student, enabling one-step generation with reasonable quality, and outperforming the teacher model with about 4 steps. DMD [26] proposes to combine a distribution matching objective and a regression loss to distill a one-step generator. The recent SDXL-Lightning [29] combines progressive distillation with adversarial loss to mitigate the blurry generation issue and ease the convergence of multi-step settings. In addition, SDXL-Lightning refines the design of the discriminator and proposes two adversarial loss objectives to balance sample quality and mode convergence.

When it comes to video models, VideoLCM [40] and AnimateLCM [21] adopt consistency distillation to enable 4-step generation with comparable quality to the full-step pre-trained video diffusion model. However, in the one-step setting, there are still considerable performance gaps observed for the visual quality. Animate-Diff Lightning [22] incorporates adversarial distillation to further reduce warps and blurs in the 1-2 step setting, despite that the model still underperforms full-step baselines.

## 3 Method

Our goal is to generate high-fidelity and temporally consistent videos in as few sampling steps as possible (*i.e.*, 1 step). The adversarial objective has been proven effective in reducing the number of sampling steps required by diffusion models in image space [27, 28, 25, 51]. However, limited efforts have been conducted on scaling up the effective adversarial training to reduce the number of sampling steps for video diffusion models. In the following, we introduce the framework of latent adversarial training to obtain efficient video diffusion model by running sampling in *single* step. In this framework, we initialize the generator and part of the discriminator with the weights of a pre-trained video diffusion model. Moreover, we introduce a structure with separate spatial and temporal discriminator heads to enhance frame quality and motion consistency.

### 3.1 Preliminaries of Stable Video Diffusion

Our method is built upon the Stable Video Diffusion (SVD) [13], which is an implementation of the EDM-framework [33] for conditional video generation, where the diffusion process is conducted in latent space. We choose the *publicly released* image-to-video generation pipeline of SVD due to its superior performance in generating high-quality and motion-consistent videos.

**Training Diffusion Models with EDM.** To facilitate the presentation, let $p_{data}(x_0)$ denote the data distribution and $p(x; \sigma)$ represent the distribution obtained by adding $\sigma^2$-variance Gaussian noise to the data. For sufficiently large $\sigma_{max}$, $p(x; \sigma_{max}) \approx \mathcal{N}(0, \sigma_{max}^2)$. Starting from high variance Gaussian noise $x_M \sim \mathcal{N}(0, \sigma_{max}^2)$, the diffusion models sequentially denoise towards $\sigma_0 = 0$ through the numerical simulation of the *Probability Flow* ODE [52]. The denoiser, $D_\theta$, attempts to predict the clean $x_0$ and is trained via denoising score matching:

$$\mathbb{E}_{x_0 \sim p_{data}(x_0), (\sigma, \mathbf{n}) \sim p(\sigma, \mathbf{n})} \left[ \lambda_\sigma \| D_\theta(x_0 + \mathbf{n}; \sigma) - x_0 \|_2^2 \right], \tag{1}$$

where $p(\sigma, \mathbf{n}) = p(\sigma) \mathcal{N}(\mathbf{n}; 0, \sigma^2)$, $p(\sigma)$ is a distribution over noise levels $\sigma$, and $\lambda_\sigma : \mathbb{R}^+ \to \mathbb{R}^+$ is a weighting function.

EDM [33] parameterizes the denoiser $D_\theta$ as:

$$D_\theta(x; \sigma) = c_{skip}(\sigma)x + c_{out}(\sigma)F_\theta(c_{in}(\sigma)x; c_{noise}(\sigma)), \tag{2}$$

where $F_\theta$ is the network to be trained. The preconditioning functions are set as $c_{skip}(\sigma) = (\sigma^2 + 1)^{-1}$, $c_{out}(\sigma) = \frac{-\sigma}{\sqrt{\sigma^2 + 1}}$, $c_{in}(\sigma) = \frac{1}{\sqrt{\sigma^2 + 1}}$, and $c_{noise}(\sigma) = 0.25 \log \sigma$.

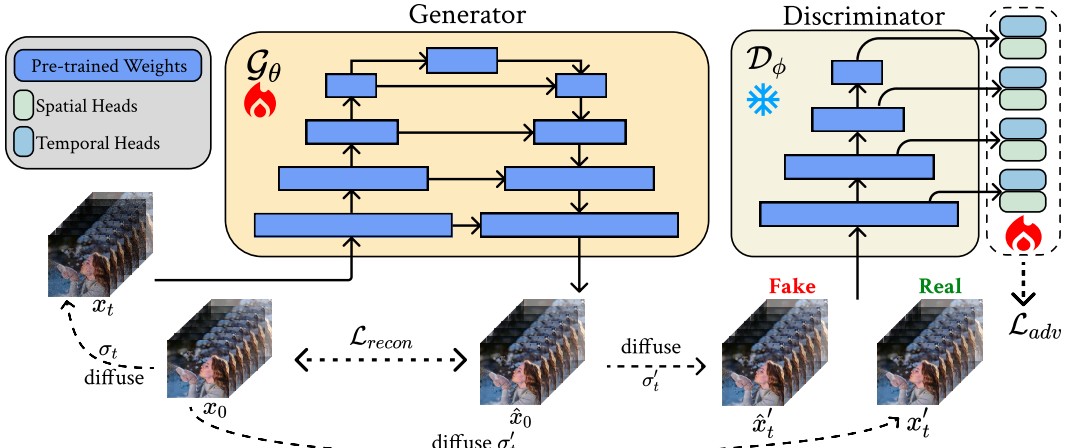

Figure 2: **Training Pipeline.** We initialize our generator and discriminator using the weights of a pre-trained image-to-video diffusion model. The discriminator utilizes the encoder part of the UNet as its backbone, which remains *frozen* during training. We add a spatial discriminator head and a temporal discriminator head after each downsampling block of the discriminator backbone and only update the parameters of these heads during training. Given a video latent $x_0$, we first add noise $\sigma_t$ through a forward diffusion process to obtain $x_t$. The generator then predicts $\hat{x}_0$ given $x_t$. We calculate the reconstruction loss $\mathcal{L}_{recon}$ between $x_0$ and $\hat{x}_0$. Additionally, we add noise level $\sigma'_t$ to both $x_0$ and $\hat{x}_0$ to obtain real and fake samples, $x'_t$ and $\hat{x}'_t$. The adversarial loss $\mathcal{L}_{adv}$ is then calculated using these real and fake sample pairs.

**Stable Video Diffusion.** The training of video model asks for a dataset of videos, each consisting of $N$ frames with height $H$ and width $W$. Given a video $\mathbf{V}_0 = \{\mathbf{I}_0^i\}_{i=0}^N$, where $\mathbf{I}_0^i \in \mathbb{R}^{3 \times H \times W}$, SVD [13] maps each frame separately to latent space using a frame encoder, $E$. The encoded frames are represented as $x_0 = \{E(\mathbf{I}_0^i)\}_{i=0}^N$, resulting in $x_0 \in \mathbb{R}^{N \times 4 \times \tilde{H} \times \tilde{W}}$. Here, $x_0 \sim p_{data}(x_0)$ is a sequence of $N$ latent frames with 4 channels, height $\tilde{H}$, and width $\tilde{W}$.

SVD inflates a text-to-image diffusion model to a text-to-video diffusion model [10]. The text conditioning is replaced with image conditioning to create an image-to-video diffusion model. Consequently, the parameterized denoiser $D_\theta$ in Eq. (2) is modified as follows:

$$D_\theta(x; \sigma, \mathbf{c}) = c_{skip}(\sigma)x + c_{out}(\sigma)F_\theta(c_{in}(\sigma)x; c_{noise}(\sigma), \mathbf{c}), \tag{3}$$

where $\mathbf{c}$ is the image condition $\mathbf{I}_0^0$, *i.e.*, the first frame of the video.

At sampling time, $D_\theta$ is leveraged to restore $x_{t-1}$ from $x_t$ using the following relation [33]:

$$d_t = (x_t - D_\theta(x_t; \sigma_t, \mathbf{c}))/\sigma_t; \quad x_{t-1} = x_t + (\sigma_{t-1} - \sigma_t) \cdot d_t, \tag{4}$$

where $\sigma_t$ is obtained with

$$\sigma_t = (\sigma_{\min}^{1/\rho} + \frac{t}{T-1}(\sigma_{\max}^{1/\rho} - \sigma_{\min}^{1/\rho}))^\rho, \tag{5}$$

where $T$ is the total number of denoising steps and $\rho$ is a hyper-parameter controlling the emphasis level to low noise levels.

### 3.2 Latent Adversarial Training for Video Diffusion Model

**Design of Networks.** Diffusion-GAN hybrid models are designed for training with large denoising step sizes [25, 27, 28, 51]. Our training procedure, illustrated in Fig. 2, involves two networks: a generator $\mathcal{G}_\theta$ and a discriminator $\mathcal{D}_\phi$. The generator is initialized from a pre-trained UNet diffusion model with weights $\theta$ (*i.e.*, the UNet from SVD). The discriminator is *partially* initialized from a pre-trained UNet diffusion model. Namely, the backbone of the discriminator shares the same architecture and weights as the pre-trained UNet encoder, and the weights of this backbone are kept frozen during training. Additionally, we *augment* the discriminator by adding a spatial discriminator head and

a temporal discriminator head after each backbone block. Therefore, in total, the discriminator comprises four spatial discriminator heads and four temporal discriminator heads. Only the parameters in these heads are trained during the discriminator training steps. The detailed architecture of these heads will be further discussed in Sec. 3.3.

**Latent Adversarial Training.** We use a pair of generated samples $\hat{x}_0$ and real samples $x_0$ to conduct the adversarial training. Specifically, during training, the generator $\mathcal{G}_\theta$ produces *generated* samples $\hat{x}_0(x_t; \sigma_t, \mathbf{c})$ from noisy data $x_t$. The noisy data points are derived from a dataset of *real* latents $x_0$ via a forward diffusion process $x_t = x_0 + \sigma_t \epsilon$. We sample $\sigma_t$ uniformly from the set $\{\sigma_1, \cdots, \sigma_{T_g-1}\}$, obtained by setting $T$ to $T_g$ and $t \in \{1, 2, \cdots, T_g - 1\}$ in Eq. (5). In practice, we set $T_g = 4$. The generated sample $\hat{x}_0$ is given by:

$$\hat{x}_0(x_t; \sigma_t, \mathbf{c}) = c_{skip}(\sigma_t)x_t + c_{out}(\sigma_t)\mathcal{G}_\theta(c_{in}(\sigma_t)x_t; c_{noise}(\sigma_t), \mathbf{c}). \tag{6}$$

To train the discriminator, we forward the generated samples $\hat{x}_0$ and real samples $x_0$ into it, aiming to let the discriminator distinguish between them. However, for a more stabilized training, inspired by exiting works [28], we add noise to the samples before passing them to the discriminator, since the backbone of the discriminator is initialized from a pre-trained UNet with weights frozen during training. Namely, we sample $\sigma'_t$ from the set $\{\sigma'_1, \cdots, \sigma'_{T_d-1}\}$, obtained by setting $T$ to $T_d$ and $t \in \{1, 2, \cdots, T_d - 1\}$ in Eq. (5), according to a discretized lognormal distribution defined as:

$$p(\sigma'_t) \propto erf\left(\frac{\log(\sigma'_t - P_{mean})}{\sqrt{2}P_{std}}\right) - erf\left(\frac{\log(\sigma'_{t-1} - P_{mean})}{\sqrt{2}P_{std}}\right), \tag{7}$$

where $P_{mean}$ and $P_{std}$ control the noise level added to the samples before passing them to the discriminator. A visualization of how different $P_{mean}$ and $P_{std}$ affect the probability of $\sigma'$ sampled is illustrated in Fig. 6. In practice, we set $T_d = 1,000$. We diffuse the real and generated samples through the forward process to obtain $\hat{x}'_t = \hat{x}_0 + \sigma'_t\epsilon$ and $x'_t = x_0 + \sigma'_t\epsilon$, respectively.

Following literature [27, 53, 54], we use the hinge loss [55] as the adversarial objective function for improved performance. The adversarial optimization for the generator $\mathcal{L}^{\mathcal{G}}_{adv}(\hat{x}_0, \phi)$ is defined as:

$$\mathcal{L}^{\mathcal{G}}_{adv} = \mathbb{E}_{\sigma,\sigma',x_0}[\mathcal{D}_\phi(c_{in}(\sigma'_t)\hat{x}'_t)], \tag{8}$$

Furthermore, we notice that a reconstruction objective, $\mathcal{L}_{recon}$, between $x_0$ and $\hat{x}_0$ can significantly improve the stability of the training process. We use Pseudo-Huber metric [56, 43] for reconstruction loss, as:

$$\mathcal{L}_{recon}(\hat{x}_0, x_0) = \sqrt{\|\hat{x}_0 - x_0\|_2^2 + c^2} - c, \tag{9}$$

where $c > 0$ is an adjustable constant. Thus, the overall objective for training the generator is as follows with $\lambda$ balances two losses:

$$\mathcal{L}^{\mathcal{G}} = \mathcal{L}^{\mathcal{G}}_{adv} + \lambda\mathcal{L}_{recon}(\hat{x}_0, x_0). \tag{10}$$

Other other hand, the discriminator is trained to minimize:

$$\mathcal{L}^{\mathcal{D}}_{adv} = \mathbb{E}_{\sigma',x_0}[\max(0, 1 + \mathcal{D}_\phi(c_{in}(\sigma'_t)x'_t)) + \gamma R1] + \mathbb{E}_{\sigma,\sigma',x_0}[\max(0, 1 - \mathcal{D}_\phi(c_{in}(\sigma'_t)\hat{x}'_t)))], \tag{11}$$

where $R1$ denotes the R1 gradient penalty [57, 27]. Here, we omit other conditional input for $\mathcal{D}_\phi$, such as $c_{noise}(\sigma')$ and image conditioning $\mathbf{c}$, for simplicity.

**Discussion.** Our latent adversarial training framework is largely inspired by LADD [28]. Similar to LADD, we set $T_g = 4$ in practice and utilize a pre-trained diffusion model as part of the discriminator. However, our approach has several key differences compared with LADD [28]. *First*, we extend the image latent adversarial distillation framework to the video domain by incorporating spatial and temporal heads to achieve one-step generation for video diffusion models. The specifics of the spatial and temporal heads are discussed in Sec. 3.3. *Second*, based on the EDM-framework [33], we observe that sampling $t'$ using a discretized lognormal distribution provides more stable adversarial training compared to the logit-normal distribution used in LADD [28]. *Finally*, unlike LADD [28], we utilize real video data instead of synthetic data for training and incorporate a reconstruction objective (*i.e.*, Eq. (9)) to ensure more stable training.

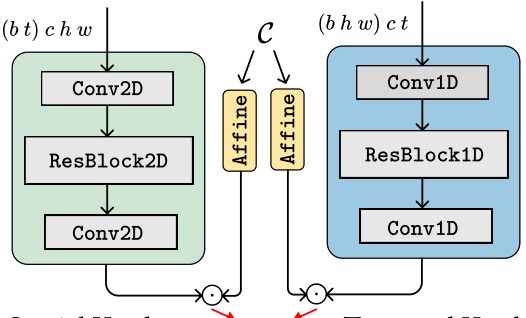

$(b\,t)\,c\,h\,w$    $\mathcal{C}$    $(b\,h\,w)\,c\,t$

Conv2D    Affine   Affine    Conv1D

ResBlock2D        ResBlock1D

Conv2D         Conv1D

Spatial Head   **Per-pixel hinge loss** Temporal Head

Figure 3: **Spatial & Temporal Discriminator Heads.** Our discriminator heads take in intermediate features of the UNet encoder. Follow exiting arts [54, 53], we use image conditioning and frame index as the projected condition **c**. **Left:** For spatial discriminator heads, the input features are reshaped to merge the temporal axis and the batch axis, such that each frame is considered as an independent sample. **Right:** For temporal discriminator heads, we merge spatial dimensions to batch axis.

Table 1: **Comparison Results.** We compare our method against SVD [13], AnimateLCM [21], UFOGen [25], and LADD [28] using different numbers of sampling steps. AnimateLCM* indicates the usage of the officially provided 25-frame model, with only the first 14 frames considered for FVD calculation. $^\dagger$ indicates our implementations. We also report the latency of the denoising process for each setting, measured on a single NVIDIA A100 GPU.

| Name | FVD↓ | Steps | Latency (s) |
|---|---|---|---|
| SVD [13] | 153.4 | 25 | 10.79 |
| | 194.4 | 16 | 6.89 |
| | 488.6 | 8 | 3.44 |
| | 1687.0 | 4 | 1.72 |
| AnimateLCM* [21] | 321.1 | 8 | 3.25 |
| | 403.2 | 4 | 1.62 |
| | 521.9 | 2 | 0.82 |
| AnimateLCM [21] | 281.0 | 8 | 1.85 |
| | 801.4 | 4 | 0.92 |
| | 1158.4 | 2 | 0.46 |
| UFOGen$^\dagger$ [25] | 1917.2 | 1 | 0.30 |
| LADD$^\dagger$ [28] | 1893.8 | 1 | 0.30 |
| Ours | 180.9 | 1 | 0.30 |

### 3.3 Spatial Temporal Heads

To train the discriminator for better understanding of the spatial information and temporal correlation, we employ separate spatial and temporal discriminator heads for adversarial training [31, 32]. The backbone of the discriminator is the encoder from the pre-trained diffusion model (*i.e.*, UNet), which consists of four spatial-temporal blocks sequentially [10]. The first three blocks downsample the spatial resolution by a factor of 2, and the last block maintains the spatial resolution. We extract the output features from each spatial-temporal block and utilize a spatial head and a temporal head to determine whether the sample is real or fake. The discriminator can be conditioned on additional information via projection [58] to enhance performance. In our setting, we use the image condition **c** and $\sigma'$ as the projected condition $\mathcal{C}$.

**Spatial Head.** For an input feature of shape $b \times t \times c \times h \times w$, the spatial discriminator first reshapes it to $(bt) \times c \times h \times w$. This way, each frame feature in a video is processed separately. The architecture for our proposed spatial head is illustrated in the left part of Fig. 3.

**Temporal Head.** Even though the features obtained from the discriminator backbone contain spatial-temporal information, we observe that using only spatial discriminator heads causes the generator to produce frames that are all identical to the image condition. To achieve better temporal performance (*e.g.*, more vivid motion), we propose to add a temporal discriminator head parallel to the spatial discriminator head. The input features are reshaped to $(bhw) \times c \times t$ instead. The architecture for our temporal head is illustrated in the right part of Fig. 3.

## 4 Experiment

**Implementation Details.** We apply Stable Video Diffusion [13] as the base model across our experiments. All the experiments are conducted on an internal video dataset with around one million videos. We fix the resolution of the training videos as $768 \times 448$ with the FPS as 7. The training is conducted for 50K iterations on 8 NVIDIA A100 GPUs, using the SM3 optimizer [59] with a learning rate of $1e-5$ for the generator (*i.e.*, UNet) and $1e-4$ for the discriminator. We set the momentum and $\beta$ for both optimizers as 0.5 and 0.999, respectively. The total batch size is set as 32 using a 4

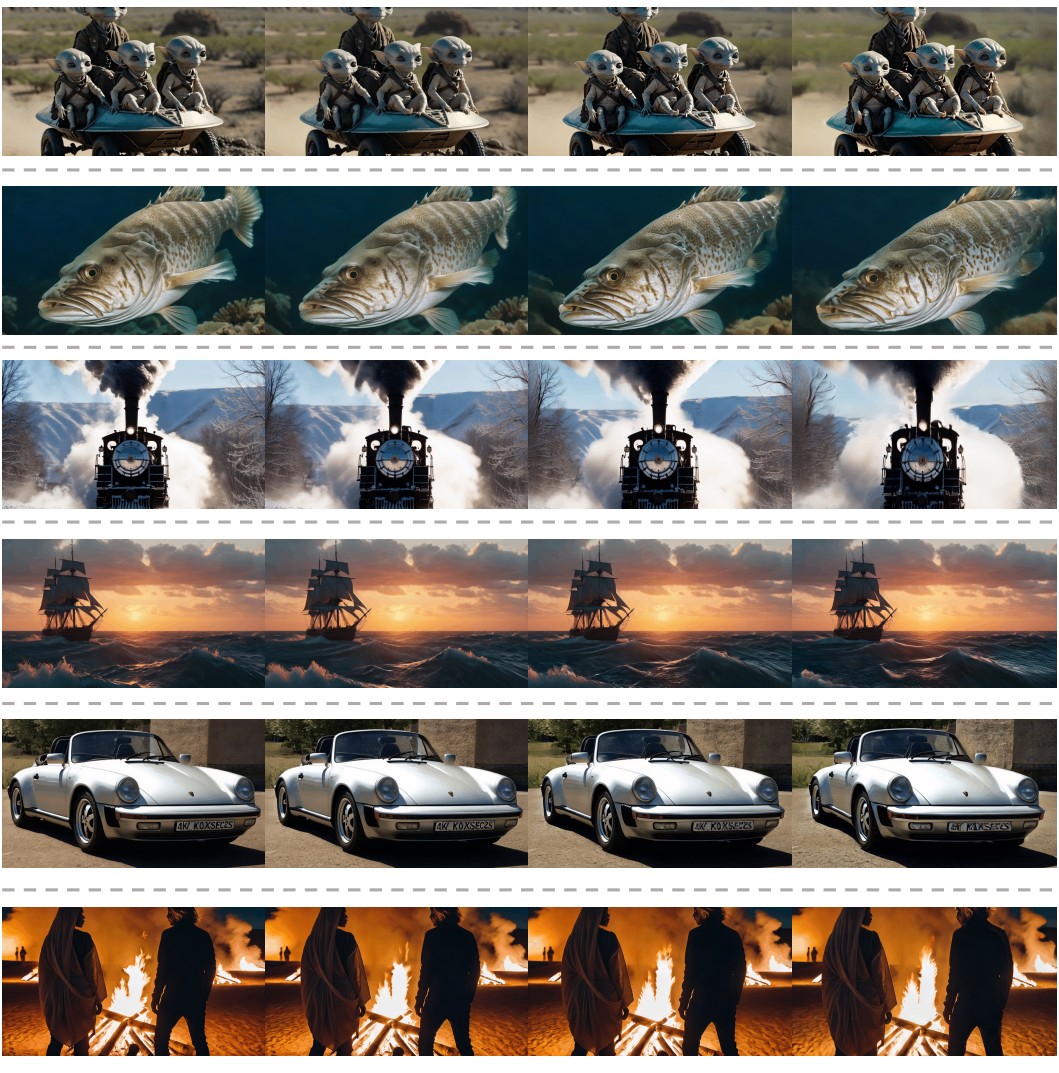

Figure 4: **Video Generation on Single Conditioning Images from Various Domains.** We employ our method on various images generated by SDXL [60] to synthesized videos. The videos contain 14-frame at a resolution of $1024 \times 576$ with 7 FPS. The results demonstrate that our model can generate high-quality motion-consistent videos of various objects across different domains. Please refer to our webpage for whole video sequences.

steps gradient accumulation. We set the EMA rate as 0.95. We set $P_{mean} = -1, P_{std} = -1$, and $\lambda = 0.1$ if not otherwise noted. At inference time, we sample videos at resolution of $1024 \times 576$.

## 4.1 Qualitative Visualization

To comprehensively evaluate the capabilities of our method, we use SDXL [60] (with refiner) to generate images of different scenes at the resolution of $1024 \times 1024$. We then perform center crop on the generated images to get resolution as $1024 \times 576$, which serves as the condition of our approach to synthesize videos of 14 frames at 7 FPS. As shown in Fig. 4, our method can successfully generate videos of high-quality frames and consistent object movements with *only* 1 step during inference.

## 4.2 Comparisons Results

**Quantitative Comparisons.** We present a comprehensive evaluation of our method compared to the existing state-of-the-art approach, AnimateLCM [21], UFOGen [25], LADD [28], and SVD [13]. To

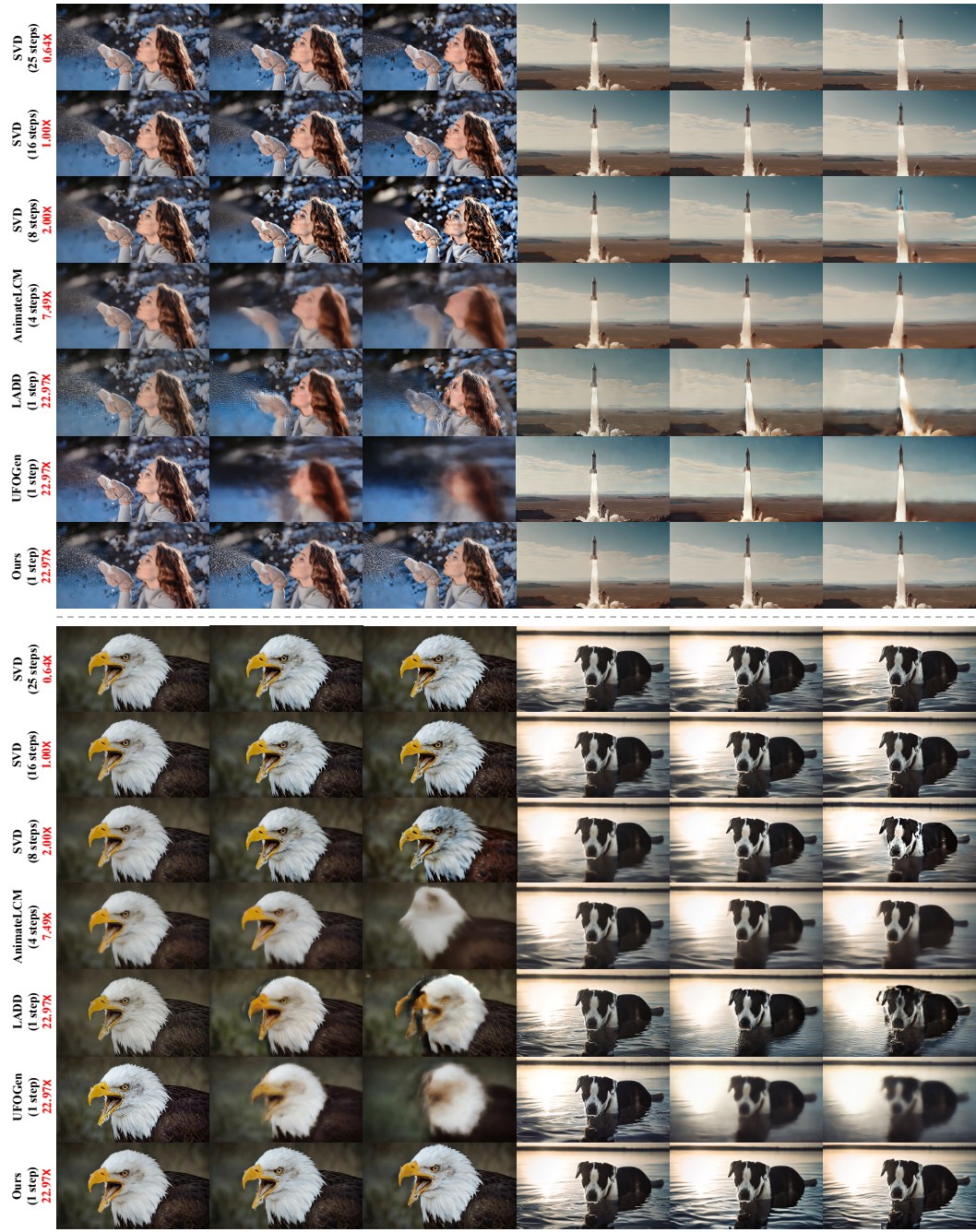

Figure 5: **Comparison between SVD [13], AnimateLCM [21], LADD [28], UFOGen [25], and Our Approach.** We provide the synthesized videos (sampled frames) under various settings for different approaches. We use SVD to generate videos under 25, 16, and 8 sampling steps, AnimateLCM to synthesize videos under 4 sampling steps, LADD and UFOGen to generate videos under 1 sampling step. AnimateLCM, LADD and UFOGen generates blurry frames with few-steps and single-step sampling. Our approach can accelerate the sampling speed by 22.9× compared with SVD while maintaining similar frame quality and motion consistency.

conduct a fair comparison on the SVD model, we train the AnimateLCM, UFOGen, and LADD on SVD using our video dataset. We follow the released code and instructions provided by AnimateLCM authors. Additionally, we include the officially released AnimateLCM-xt1.1 [21] by evaluating the first 14 generated frames and denote the approach as AnimateLCM*. We try our best to implement LADD [28] and UFOGen [25] and denote respectively as LADD†, and UFOGen†. Note that simply re-using the discriminator from LADD [28] and UFOGen [25] leads to *out-of-memory issue*, since the computation in the video model is much larger than the image model. Here we replace the discriminator from LADD [28] and UFOGen [25] with the one proposed in our work.

We follow exiting works [61] by using Fréchet Video Distance (FVD) [30] as the comparison metric. Specifically, we use the first frame from the UCF-101 dataset [62] as the conditioning input and generate 14-frame videos at a resolution of $1024 \times 576$ at 7 FPS for all methods. The generation results are then resized back to $320 \times 240$ for FVD calculation. Our method is compared against SVD [13] and AnimateLCM [21], each using a different number of sampling steps. Furthermore, to better demonstrate the effectiveness of our method, we measure the generation latency of each method, which is calculated on running the diffusion model (*i.e.*, UNet). Note that only for SVD [13], classifier-free guidance [63] is used, leading to higher computational cost.

As shown in Tab. 1, our method achieves comparable results to the base model using 16 discrete sampling steps, resulting in approximately a $23\times$ speedup. Our method also outperforms the 8-steps sampling results for AnimateLCM and AnimateLCM*, indicating a speedup of more than $6\times$. For *single-step* evaluation, our method performs much better than existing step-distillation methods [25, 28] built upon image-based-diffusion models.

**Qualitative Comparisons.** We further provide qualitative comparisons across different approaches by using publicly available web images. Fig. 5 presents generation results from SVD [13] with 25, 16, and 8 sampling steps, AnimateLCM [21] with 4 sampling steps, UFOGen [25], LADD [28], and our method with 1 sampling step. As can be seen, our method achieves results comparable to the sampling results of SVD using 16 or 25 denoising steps. We notice significant artifacts for videos synthesized by SVD when using 8 denoising steps. Compared to AnimateLCM [21],UFOGen [25], and LADD [28], our method produces frames of higher quality and better temporal consistency, with fewer or same denoising steps, demonstrating the effectiveness of our proposed approach.

## 4.3 Ablation Analysis

**Effect of Discriminator Heads.** We explore the effect of our proposed spatial and temporal heads by measuring the FVD on the UCF-101 dataset. We conduct latent adversarial training with three different discriminator settings to analyze the impact of our spatial and temporal discriminators. As shown in Tab. 2, training with only spatial heads (denoted as *SP*) or only temporal heads (denoted as *TE*) results in significantly worse performance than using all of them (denoted as *SP+TE*).

Nevertheless, since our discriminator backbone shares the same architecture as the spatial-temporal generator, the receptive field of each pixel on the feature maps provided by the backbone can cover a region both spatially and temporally. Additionally, we embed the frame index as an additional projected condition. Consequently, even when using only spatial heads or only temporal heads, the generated videos still exhibit reasonable frame quality and temporal coherence.

**Effect of Noise Distribution for Discriminator.** As shown in Fig. 6, following Eq. (5), $P_{mean}$ and $P_{std}$ control the distribution of $\sigma'_t$, which is the noise level added to $x_0$ or $\hat{x}_0$ before passing to the discriminator as real and fake samples, respectively. We explore the effect of different noise distributions on model performance by calculating FVD on the UCF-101 dataset.

When the sampled $\sigma'_t$ is concentrated on small values, *e.g.*, $P_{mean} = -2$ and $P_{std} = -1$ in our case, we notice that the discriminator can quickly learn to distinguish real samples from fake ones. This leads to a significant drop in performance, as shown in Tab. 3 and Fig. 7.

On the other hand, when the noise level becomes too high, *e.g.*, $P_{mean} = 1$ and $P_{std} = 1$, the discriminator input, which is $c_{\text{in}}(\sigma'_t)\hat{x}'_t = \frac{\hat{x}_0 + \sigma'_t \epsilon}{\sqrt{\sigma'^2_t + 1}}$, results in small adversarial gradients for the generator. This causes increased artifacts in the generated videos, as shown in Fig. 7 and Tab. 3.

Table 2: **Analysis of discriminator.** We measure FVD for models with different discriminator configurations. "SP" indicates that spatial heads and "TE" indicates temporal heads.

| | SP+TE | SP | TE |
|---|---|---|---|
| FVD | **180.9** | 514.7 | 539.2 |

Table 3: **FVD vs. $\sigma'$ distributions.**

| $P_{mean}$ | $P_{std}$ | FVD |
|---|---|---|
| $-2.0$ | $-1.0$ | 3370.4 |
| $-1.0$ | $-1.0$ | **180.9** |
| 0.0 | 1.0 | 416.7 |
| 1.0 | 1.0 | 632.9 |

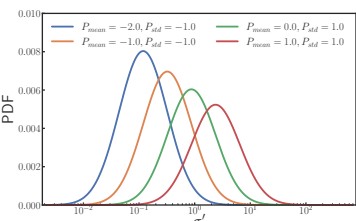

Figure 6: **PDF of $\sigma'$.**

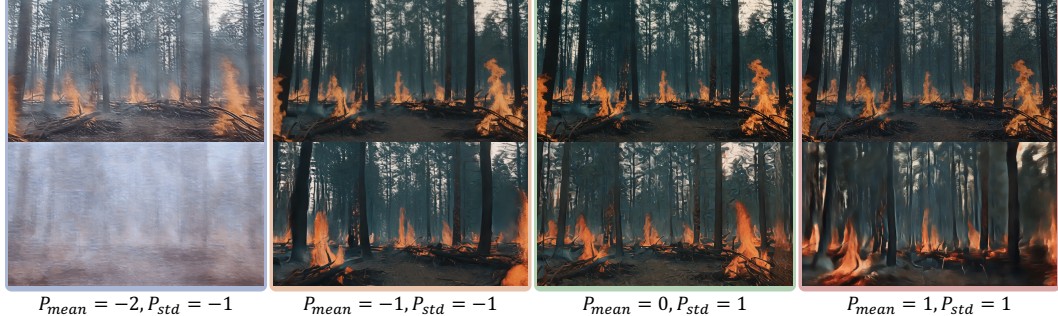

| $P_{mean} = -2, P_{std} = -1$ | $P_{mean} = -1, P_{std} = -1$ | $P_{mean} = 0, P_{std} = 1$ | $P_{mean} = 1, P_{std} = 1$ |

Figure 7: **Analysis of $\sigma'$ Distributions.** We investigate the impact of changing the distribution of $\sigma'$ by adjusting $P_{mean}$ and $P_{std}$. The results are shown with the same image conditioning. The first row and the second row display the first and last frames generated, respectively.

## 5  Discussion and Conclusion

In this work, we leverage adversarial training to reduce the denoising steps of the video diffusion model and thus improve its generation speed. We further enhance the discriminator by introducing spatial-temporal heads, resulting in better video quality and motion diversity. We are the first to achieve *1-step* generation for video diffusion models while preserving comparable visual quality and FVD scores, democratizing efficient video generation to a broader audience by delivering more than $20\times$ speedup for the denosing process.

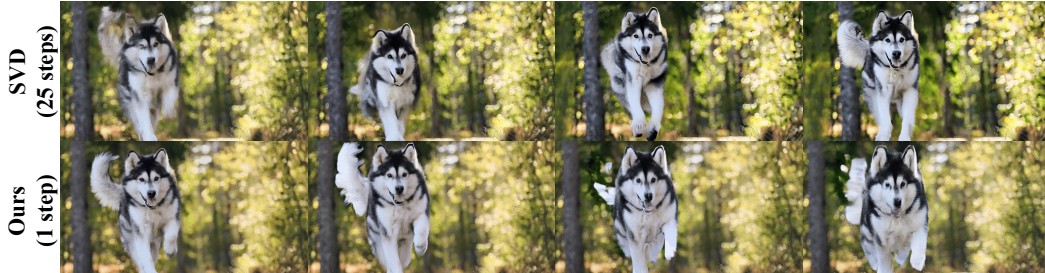

Figure 8: **Limitations.** We show that, for some conditional images, our model tends to generate a few unsatisfactory frames when complex motion might be required (*Second Row*). Similar artifacts can also be observed in frames generated from SVD by sampling at 25-steps (*First Row*).

**Limitations.** We observe that when the given conditioning image indicates complex motion, *e.g.* running, our model tends to generate unsatisfactory results, *e.g.* blurry frames, as shown in Fig. 8. Such artifacts are introduced by the original SVD model, as can be observed in Fig. 8. We believe a better text-to-video model can solve such issue.

This work successfully achieves *single* sampling step for video diffusion models. However, under such setting, the temporal VAE decoder and the encoder for image conditioning take a considerable portion of the overall runtime. We leave the acceleration of these models as future work.

## Acknowledgments and Disclosure of Funding

This research has been partially funded by grants to D. Metaxas from NSF: 2310966, 2235405, 2212301, 2003874, 1951890, AFOSR 23RT0630, and NIH 2R01HL127661.

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
