# OpenReview forum: "SF-V: Single Forward Video Generation Model"
_NeurIPS.cc/2024/Conference — NeurIPS 2024 poster_

### Official Review · Reviewer_Q9Lc · 2024-07-10

**Soundness:** 3
**Presentation:** 3
**Contribution:** 3
**Rating:** 5
**Confidence:** 5

**Summary:**

This paper proposes a method to accelerate video generation model inference speed by distilling the multi-step reasoning of Singular Value Decomposition (SVD) into a single-step generation using adversarial networks. This approach achieves comparable results to multi-step SVD generation while significantly improving inference efficiency. Experimental results validate the effectiveness of the authors' method.

**Strengths:**

1. The algorithm proposed by the authors significantly improves video generation inference speed without compromising the quality of the generated results.
2. The authors propose using two parallel heads, a spatial head and a temporal head, to implement the discriminator. This resolves the issue of a single head potentially leading to the generation of static videos.

**Weaknesses:**

1. The authors trained the entire network using 1 million internal videos. For those without access to this dataset, reproducing this work could be challenging. Will the authors make these data and source code publicly available?
2. How many training data points are needed to ensure that the distilled model has strong generalization capabilities? The training process for SVD likely requires far more than 1 million data. Is using 1 million data for distillation training sufficient?
3. The author mentioned using the encoder of a Unet for initializing the discriminator. What is the insight behind this choice? Why did they consider this initialization method?
4. In Adversarial Diffusion Distillation, they use distillation loss to transfer knowledge from the teacher model to the student model. Why did the authors not use this approach here? Is there any specific reason?

**Questions:**

see Weaknesses

**Limitations:**

It's unclear how well the method generalizes currently, as the authors only conducted quantitative analysis on the UCF101 dataset. This uncertainty arises because the authors trained the model using only 1 million data points.

---

> ### Author Rebuttal · Authors · 2024-08-05
>
> We thank the reviewer for the constructive comments. The detailed responses regarding each concern are listed below.
>
> ---
>
> **Q1. Reproducing the results.**
>
> A1. We plan to make our code and checkpoint public for the reproducing of our work.
>
> ---
>
> **Q2. Training data.**
>
> A2. Our dataset contains around 1M video clips of various lengths. As we mentioned in the paper (Line 191-192), our model is trained for 50K iterations with a total batch size of 32. For each training sample, we randomly select 14 consecutive frames for a video temporally downsampled to 7 FPS. We agree our dataset is far less the one used for SVD training. However, in this work, we fine-tune SVD for efficient generation instead of learning motion from scratch, which makes our model requires less data. We humbly believe it is an advantage of our approach that we can require much less data than SVD while being able to fine-tune the model to single-step.
>
> On the other hand, we do expect our model can benefit from more high-quality data training and longer training iterations. To validate the assumption, we run the following experiment.
>
> - First, we randomly selected 700K videos out of the 1M dataset and use that to train our model.  Our method achieves FVD of $275.7$ with 700K videos while $180.9$ using 1M videos. We also provide the qualitative results in Fig. C of the *attached one-page PDF*.
>
> - Second, we report the performance of the model under various training iterations. Due to the limit of computation, we report results for training the model within 50K iterations. In the following table, we show how the FVD score changes along the training process.
>
> >| Iters (K) | 20 | 30 | 40 | 50 |
> | --- | --- | --- | --- | --- |
> | FVD | 542.2 | 479.5 | 217.1 | 180.9 |
>
> ---
>
> **Q3. Discriminator initialization.**
>
> A3. We use the encoder of the UNet to initialize the backbone for its strong video understanding ability and we fix the weights of the backbone during training. Such design can further facilitate faster training and fewer memory consumption.
>
> To further demonstrate the effectiveness, we train the discriminator backbone together with the added heads *without* initializing it from the pre-trained UNet encoder. We observe a significant drop of performance, *i.e.,* the FVD on UCF-101 is 1490.2 (without initialization from UNet) vs. 180.9 (with initialization of UNet.). We also provide qualitative results in Fig. C of the *attached one-page-PDF*.
>
> ---
>
> **Q4. Distillation loss.**
>
> A4. There are two main reasons for not using the teacher-student distillation loss in ADD [26].
>
> - First, to obtain distillation target in ADD [26], during training, the denoising process for the teacher model is required. Thus, the teacher model needs to run multiple inference steps. Such a training process can drastically increase the training time.
>
> - Second, distillation loss in ADD [26] needs to be calculated in the pixel space instead of latent space. Decoding the latents and calculating the gradients with the video VAE decoder will introduce significant computation overhead for video models, leading to out-of-memory issue for SVD.  Thus, we use the reconstruction objective instead of the distillation loss in our approach to make the model training feasible.
>
> ---
>
> **Q5. Generalization ability.**
>
> A5. As shown in the Supplementary Materials, we qualitatively evaluate our method on images of various styles and objects, demonstrating the strong generalization ability of our method for video generation.
>
> As for the quantitative evaluation metric, we chose FVD on UCF-101 as it is the most adopted benchmark for video generation. For example, it is adopted in SVD[14] and AnimateLCM [21]. We fully agree with the reviewer that more benchmark datasets are helpful for video model evaluation and it is an exciting direction for future work.

---

> > ### Author Response · Authors · 2024-08-09
> >
> > Dear Reviewer Q9Lc,
> >
> > Thank you for your valuable feedback on our submission.
> >
> > We have provided additional explanations to address the key points you raised, including reproducing our results, training data, effectiveness of our discriminator initialization, reasons why we don’t use distillation loss, and the generalization ability of our method. As the deadline for the Author-Reviewer discussion is approaching, we would like to kindly ask if our responses sufficiently clarify your concerns or if there are any remaining issues you would like us to address. We appreciate your time and consideration.
> >
> > Best,
> >
> > Authors

---

> > ### Comment · Reviewer_Q9Lc · 2024-08-12
> >
> > Thank you for the thorough response. The rebuttal has addressed most of my concerns. I am happy to keep my initial score.

---

> > > ### Author Response · Authors · 2024-08-12
> > >
> > > Dear Reviewer Q9Lc,
> > >
> > > Thank you for reading our rebuttal and providing a response! If there are any remaining issues you would like us to address, we are always willing to provide more clarification.
> > >
> > > Best,
> > >
> > > Authors

---

### Official Review · Reviewer_FT7s · 2024-07-11

**Soundness:** 3
**Presentation:** 4
**Contribution:** 2
**Rating:** 5
**Confidence:** 4

**Summary:**

The paper tackles the task of distilling diffusion-based text-to-video models into single-step models, achieving much higher sampling speeds.  To this end, they build a framework to fine-tune a pretrained video diffusion model. This fine-tuning is done in an adversarial setting in the latent space, whereby the generator and discriminator are designed to achieve higher image quality and video consistency. In particular, the generator is initialized from a pretrained video model, and the discriminator is partially initialized with the layers of a pretrained encoder, where additional spatial-temporal layers are added and trained. The model achieves SOTA SVD quality in the setting of 1-step video diffusion and an additional speedup compared to existing art.

**Strengths:**

The paper is very well written and easy to read. In particular, the introduction and related work well put the work in context and the contribution. The related work section is sufficiently extensive and provides the right context to identify the existing gap the paper is trying to solve. The results and visualizations are clear, well-explained, and detailed.

The paper is also well validated, showing excellent results compared to the state-of-the-art in single-step video generation, achieving superiority in terms of quality as well as sampling time.

By fine-tuning an existing video model, the authors can also achieve good training time and use any knowledge captured by the pretrained model. In that sense, the proposed approach makes sense and is well-suited for the task.

**Weaknesses:**

My main concerns are concerning the novelty and significance of the proposed approach in the following sense:

1. One component of the method is a re-design of the generator and discriminator. First, this in itself seems to be of limited novelty. The proposed components are standard and well-established in the community. Finetnuning the discriminator by adding additional layers is not new, and similarly, for the generator, copying the pretrained encoder seems sensible but does not offer a new contribution. Second, one could take the existing single-step adversarial text-to-image approaches and replace the generator and discriminator with the architecture proposed in this paper. The authors identify such papers well (e.g. [24, 25, 26, 27, 28]) but do not compare them to such a baseline. Hence, assessing whether the proposed architectural changes are meaningful is difficult.

2. The proposed training regime (adversarial training and reconstruction losses) is another component. On the flip side to 1, is this training regime important? In that sense, one could take existing single or multi-step video diffusion baselines (e.g., [14, 21]) and change the training regime to that proposed in this paper. Further, assuming that this training regime improves in the video setting, one could assume that it is more general and applicable also in the single-step text-to-image regime.

Further, the proposed components both seem to be marginally novel. Adversarial training for single-step distillation and the temporal components are also not new. Is there something non-trivial in the combination of both components?

**Questions:**

I believe the paper provides a great engineering effort, significantly improving sampling speed and quality compared to the state-of-the-art. However, I am concerned about the novelty of the proposed approach, as indicated in the weaknesses section.

**Limitations:**

Limitations are briefly addressed but not sufficiently analyzed. It would be great to see examples of errors introduced by the approach or some further analysis.

---

> ### Author Rebuttal · Authors · 2024-08-05
>
> We thank the reviewer for the constructive comments. We provide detailed response in the following.
>
> ---
>
> **Q1. About novelty.**
>
> A1. We agree with the reviewer that certain design components, *e.g.,* adversarial training and constructing the discriminator initialized from the diffusion model, have been explored in different manners in other tasks. However, we would also like to kindly emphasize that distilling a video diffusion model into single-step with the similar performance as 25-steps *has not been studied* in the literature.
>
> In the following, we would like to kindly clarify that the novelty comes from the design of the whole training pipeline that is tailored for solving a specific challenging task, instead of the individual component like adversarial training.
>
> Our framework is inspired by LADD [27], which distills an image diffusion model into fewer steps. Here, we highlight major differences compared with LADD.
>
> - _First_, we extend the image latent adversarial distillation from image to the video domain by incorporating spatial and temporal heads to achieve one-step generation for video diffusion models. The training computation to distill a video model is much larger than distilling an image model. Therefore, we carefully design our discriminator, which is not used in the image domain, to enable the stable training. Notably, our discriminator only uses certain part of the UNet that we find important. Such design is different from exiting arts that re-use the whole UNet. In this way, we can significantly reduce the training computation for adversarial training, and train the video model much faster and consume much less GPU memory.
>
> - _Second_, based on the EDM-framework, we observe that sampling $t^\prime$ using a discretized lognormal distribution provides more stable adversarial training compared to the logit-normal distribution used in LADD.
>
> - _Third_, unlike LADD, we utilize real video data instead of synthetic data for training, removing the storage and computation burden for calculating and saving the synthetic data.
>
> - *Fourth*, we employ reconstruction objective (Eq. 9 in the main paper) along with R1 gradient penalty (Eq. 11 in the main paper) to stablize the training of the video diffusion model.
>
> In we only focus on how we design the discriminator (*e.g.,* re-using some weights from the generator and fixing them), indeed, they do not look fancy and seem like lacking novelty. However, we humbly think it is important to look at how such design, when smoothly combined with techniques (like how to sampling the noise), can solve an important task.
>
> We hope such an explanation would make sense to the reviewer.
>
> ---
>
> **Q2. Comparison with the approaches that built for distilling image diffusion models [24, 25, 26, 27, 28].**
>
> A2. Directly taking existing step-distillation methods for distilling text-to-image models to video generation is not sufficient to achieve reasonable results. As suggested by the reviewer, we compare our approach with the image-based distillation methods.
>
> - First, DMD [25] and ADD [26] are too computationally costly to implement in video generation tasks. Both of these works require loss calculation in the pixel space instead of latent space. Forwarding and backwarding through the VAE decoder while training will introduce a huge GPU memory overhead and slow down the whole training process.
>
> - Second, since LADD [27] outperforms SDXL-Lightning [28] by significant margin [27], we try our best to implement LADD [27]. We also implement UFOGen [24].  We replace the generator in LADD and UFOGen with SVD. Please kindly notice that simply re-using the discriminator from LADD and UFOGen leads to the *out-of-memory issue*, since the computation in the video model is much larger than the image model. Here we replace the discriminator from LADD and UFOGen to the one proposed in our work.
>
> We report the FVD score on UCF-101 in the following table. Additionally, we provide qualitatively comparisons between our method and  LADD and UFOGen in the Fig. C of the *attached one-page-PDF*.
>
> >| Method | UFOGen | LADD | Ours |
> | --- | --- | --- | --- |
> | FVD | 1917.2 | 1893.8 | 180.9 |
>
> From these results, we can see that, for distilling video diffusion models, our approach performs much better than exiting step-distillation methods built upon image-based-diffusion models.
>
> ---
>
> **Q3. Importance of the proposed training regime**
>
> A3. We are not sure whether we fully understand this concern, yet, please allow us to try to answer them in three different perspectives.
>
> - First, as the experiments shown in A2, our adversarial training regime is very important, as it achieves better results than the ones proposed in LADD [27] and UFOGen [24], which are two methods for distilling image-based-diffusion models.
>
> - Second, the reviewers suggest apply our approach for two work: SVD[14] and AnimateLCM [21]. Actually, our video model is fine-tuned from SVD [14]. In Tab.1 of the main paper, we show single-step model achieves similar results as 25-steps SVD, and better results than 8-steps AnimateLCM [21].
>
> - Third, directly applying our training pipeline to the image-based-diffusion model is non-trivial. For example, It requires the changing of our discriminator, the noise sampling process, and the changes of adversarial losses. We humbly think such experiments are beyond the scope of this work.
>
> ---
>
> **Q4. Discussion about limitations**
>
> A4. Thank you for the suggestion. We observe that when the given conditioning image indicates complex motion, _e.g._ running, our model tends to generate unsatisfactory results, _e.g._ blurry frames, as shown in the Fig. D of the *attached one-page PDF*. Such artifacts are introduced by the original SVD model,  as can be observed in Fig. D of the *attached one-page PDF*. We believe a better text-to-video model can solve such issue and are interested in trying our step-distillation approach to such a model when there is an open-sourced one.

---

> > ### Author Response · Authors · 2024-08-09
> >
> > Dear Reviewer FT7s,
> >
> > Thank you for your valuable feedback on our submission.
> >
> > We have provided additional explanations to address the key points you raised, including our novelty, further comparisons, importance of our training regime, and limitations of our method. As the deadline for the Author-Reviewer discussion is approaching, we would like to kindly ask if our responses sufficiently clarify your concerns or if there are any remaining issues you would like us to address. We appreciate your time and consideration.
> >
> > Best,
> >
> > Authors

---

> > > ### Comment · Reviewer_FT7s · 2024-08-11
> > >
> > > I want to thank the authors for their rebuttal.
> > >
> > > While I still believe the novelty is limited, the additional evaluation has provided stronger evidence of the merit of the proposed approach compared to prior work.
> > >
> > > I still have some concerns about novelty, as noted in the author's answer (Q1). The authors note a number of differences from previous work LADD [27]. However, without ablating each of the mentioned components either with respect to speed or to generation quality, it is difficult to assess the importance of each component. I would appreciate it if the authors could address this.

---

> ### Author Response · Authors · 2024-08-11
>
> Dear Reviewer FT7s,
>
> Thank you for reading our rebuttal and providing response!
>
> We would like to apologize that our previous response in Q1 regarding the comparison between our approach and LADD does not contain the detailed explanation regarding the experiments that we have done. In the following, we connect our previous experiments along with the difference between our work and LADD.
>
> First, in Tab. 2 of the main paper, we conduct ablation experiments with different discriminator heads settings. We show that, using spatial and temporal heads achieves better generation quality than only using spatial head (which is the design from LADD). Additionally, our discriminator only uses certain part of the UNet that we find important, while LADD does not. In this way, we can significantly reduce the training computation for adversarial training, and train the video model much faster and consume much less GPU memory. By directly using the discriminator design from LADD, we notice the out of GPU memory issue on 80G Nvidia A100 GPU.
>
> Second, based on the EDM-framework, we observe that sampling $t^\prime$ using a discretized lognormal distribution provides more stable adversarial training. In Fig. 6 and Fig. 7 of the main paper, we show the importance of our proposed noise sampling schedule for achieving the better performance.
>
> Third, we utilize real video data instead of synthetic data for training, removing the computation burden for calculating and saving the synthetic data. For instance, generating 1M videos using SVD requires approximately 3K GPU hours.
>
> Fourth, we employ reconstruction objective (Eq. 9 in the main paper) along with R1 gradient penalty (Eq. 11 in the main paper) to stablize the training of the video diffusion model. Without R1 gradient or the reconstruction objective, we observe frequent training divergence.
>
> Besides, as mentioned in our answer for Q2, we use the proposed training regime from LADD [27] combined with the discriminator proposed in our work and conduct quantitative and qualitative comparison experiments. Our method significantly outperforms LADD [27] training regime in the video distillation task, demonstrating the effectiveness of our proposed training regime.
>
> Thanks again for the feedback from the reviewer. We will add the above discussion to the revised paper and hope it can be helpful to understand the difference between our work and LADD.  We would deeply appreciate it if the reviewer could reconsider the score and we are always willing to address any of your further concerns.
>
> Best,
>
> Authors

---

### Official Review · Reviewer_Do55 · 2024-07-12

**Soundness:** 3
**Presentation:** 3
**Contribution:** 2
**Rating:** 5
**Confidence:** 4

**Summary:**

The authors propose a method to generate similar-quality as the original video diffusion model Stable Video Diffusion (SVD) in a single feedforward pass.
To this end, they take the pre-trained SVD model and fine-tune it with a reconstruction and adversarial loss. The discriminator uses a frozen copy of the SVD encoder along with trained spatial & temporal readout heads to encourage both spatial and temporal consistency. Further, the authors add noise to the inputs of the discriminator to stability training.

The results are of slight better quality than 16 steps of SVD, and slightly worse than 25 steps of SVD.

**Strengths:**

- The paper is very clear and easy to read
- The presented approach is simple
- It works! Quality is somewhat comparable to several diffusion steps with the original model

**Weaknesses:**

- A large chunk of the methods section repeats how the original diffusion model is trained -- since this is already known information and not directly relevant to the presented method, it could be shortened or moved to the appendix
- The proposed method is fairly simple and incremental, though the results are better than existing methods

**Questions:**

- L257: what's the relative compute / time that goes into the encoder / decoder compared to the actual latent model?

**Limitations:**

The authors briefly discuss the fact that the VAE decoder & encoder for image conditioning.

It should be noted that training GANs can often come with the additional headache of instabilities.

---

> ### Author Rebuttal · Authors · 2024-08-05
>
> We thank the reviewer for the constructive comments. The detailed responses regarding each concern are listed below.
>
> ---
>
> **Q1. About improving the writing for the preliminary section.**
>
> A1. Thanks for the suggestion! We will revise the writing of the manuscript.
>
> ---
>
> **Q2. About the approach is simple.**
>
> A2. We agree with the reviewer that our approach, built upon adversarial training, is relative straightforward and simple to re-implement. On the other hand, as high-lighted by the reviewer, our approach actually works, which we believe further demonstrate the value of this paper - simple yet effective. In fact, our approach also outperforms other distillation based methods like latent consistency model.
>
> Additionally, this is the first paper showing that a single forward video diffusion model can obtain similar quality as a 25-step model. Such results and findings could inspire later research and efforts to develop more advanced approaches to further improve the generation quality.
>
> ---
>
> **Q3. Inference time for image encoding and encoding.**
>
> A3. We calculate sampling time for each component when generating $14$-frames videos at a resolution of $1024\times 576$. We use A100 GPU as the testbed to get the latency.
>
> - SVD use both CLIP image encoder and VAE encoder to encode the first frame of the video as the condition input:
>     - The runtime for CLIP image encoding is 0.03 s.
>     - The runtime for VAE image encoding is 0.01 s.
> - The latency for forwarding the UNet once is 0.30 s.
> - The latency for VAE decoding the latent to get videos is 2.32 s.
>
> ---
>
> **Q4. Training instabilities of GANs.**
>
> A4. We agree with the reviewer that applying GAN during training is non-trivial. To avoid the mode collapse and achieve the stabilized training, we carefully design our training pipeline. For instance, we use hinge loss as the adversarial training. Additionally, we employ reconstruction objective (Eq. 9 in the main paper) and R1 gradient penalty (Eq. 11 in the main paper) during training to make the training process more stable. Moreover, based on the EDM-framework, we observe that sampling $t^\prime$ using a discretized lognormal distribution provides more stable adversarial training.
>
> We have also tried to re-train our model multiple times using the same dataset and approach. The training is stable across different runs and we can re-produce very similar results.

---

> > ### Author Response · Authors · 2024-08-09
> >
> > Dear Reviewer Do55,
> >
> > Thank you for your valuable feedback on our submission.
> >
> > We have provided additional explanations to address the key points you raised, including our approach, inference time, and training instabilities. As the deadline for the Author-Reviewer discussion is approaching, we would like to kindly ask if our responses sufficiently clarify your concerns or if there are any remaining issues you would like us to address. We appreciate your time and consideration.
> >
> > Best,
> >
> > Authors

---

> > > ### Author Response · Authors · 2024-08-13
> > >
> > > Dear Reviewer Do55,
> > >
> > > We would like to thank you again for your valuable feedback on our paper.
> > >
> > > As the period for the Author-Reviewer discussion is closing today, we would like to use this opportunity to kindly ask if our responses sufficiently clarify your concerns. We sincerely appreciate your time and consideration.
> > >
> > > Best,
> > >
> > > Authors

---

### Official Review · Reviewer_R1mD · 2024-07-23

**Soundness:** 3
**Presentation:** 3
**Contribution:** 2
**Rating:** 4
**Confidence:** 5

**Summary:**

The paper proposes an idea of training a distillation approach using GAN based technique. The advantage which is suggested by the authors is that such distillation approach can reduce the computational cost associated with the sampling new samples during the inference time. Instead of taking multiple sampling steps only single step is required for inference.

**Strengths:**

The idea is sound and well founded. Utilizing GAN based distillation approach to reduce the computational overhead related to the sampling process in diffusion models. This is especially important for videos since a minute long video consists of almost 1.5k frames.

**Weaknesses:**

- Videos do not look like they're real world videos. How does this work with the real world videos.

- The motion is very laminar in the videos. How is the performance when trained on videos with non laminar motion.

**Questions:**

- GANs suffer from mode collapse problem how was it ensured that is not the case here with this approach?

- GANs suffer from the problem of instability in training how was it handled

- Why weren't standard video datasets like UCF101 used in evaluation?

**Limitations:**

GAN based models suffers from training instability because of the presence of adversarial loss.

---

> ### Author Rebuttal · Authors · 2024-08-05
>
> We thank the reviewer for the constructive comments. The detailed responses regarding each concern are listed below.
>
> ---
>
> **Q1. About generating real world videos.**
>
> A1. Thanks for the suggestions. In this work, we fine-tune SVD, which is an image-to-video model, into single sampling step. Our model can take *arbitrary* given image as input and generate the corresponded video.
>
> In the main paper (Line 221-222 and Fig. 5), we show the results of using the real images as input to generate the real world videos. The saved videos can be found in our Supplementary Materials. Additionally, we compare our generation results with other approaches.
>
> We also provide more examples of using real images as input to generate real world videos. The examples are shown in the Fig. A in the *attached one-page PDF*.
>
> ---
>
> **Q2. About some videos with laminar motion.**
>
> A2. Thank you for noticing the laminar motion for some videos. We would like to kindly mention that this paper aims to improve the sampling efficiency of the pre-trained SVD model, instead of improving the motion quality of SVD. In fact, we have shown in the main paper (Fig. 5) and the *Comparisons section* in the webpage of the Supplementary Material, the motion quality of our approach is *similar* to the motion of 25-steps of the original SVD model.
>
> In fact, we use training videos with large motion to fine-tune SVD. Some examples of training videos are shown in the Fig. B of the *attached one-page PDF*. Nevertheless, improving the motion of SVD without modify its whole architecture (*e.g.,* image-based encoder), is still challenging, which also beyonds the scope of this paper. We further provide more examples in Fig. C of the *attached one-page PDF*,  demonstrating that for the same conditioning image, running SVD with 25 steps can also generate videos with laminar motion.
>
> One reason of that SVD generates videos with laminar motion is due to the SVD can only synthesize short clips. We believe that in the future, with more open sourced video generation models synthesizing minutes-long videos, the motion can be significantly improved. By that time, we would be very interested in applying our approach to those models.
>
> ---
>
> **Q3. About the training challenges of using GAN, such as mode collapse and instability.**
>
> A3. We agree with the reviewer that applying GAN during training is non-trivial. To avoid the mode collapse and achieve the stabilized training, we carefully design our training pipeline. For instance, we use hinge loss during adversarial training. Additionally, we employ reconstruction objective (Eq. 9 in the main paper) and R1 gradient penalty (Eq. 11 in the main paper) during training to make the training process more stable. Moreover, based on the EDM-framework, we observe that sampling $t^\prime$ using a discretized lognormal distribution provides more stable adversarial training.
>
> We have also tried to re-train our model multiple times using the same dataset and approach. The training is stable across different runs and we can re-produce very similar results.
>
> ---
>
> **Q4. standard video datasets like UCF101 used in evaluation**
>
> A4. We agree with the reviewer that UCF-101 is a standard dataset that should be used in the evaluation.
>
> In the paper, we mainly use UCF-101 to obtain the quantitative results. For example, in Line 211 of main paper, we explain the details of how we use UCF-101. In Tab.1, Tab.2, and Tab.3, we show the calculated FVD on UCF-101 to compare our approach with others and ablating the design principles of our approach.

---

> > ### Author Response · Authors · 2024-08-09
> >
> > Dear Reviewer R1mD,
> >
> > Thank you for your valuable feedback on our submission.
> >
> > We have provided additional explanations to address the key points you raised, including the generation of real-world videos, laminar motion, and issues related to instability during training and evaluation metrics. As the deadline for the Author-Reviewer discussion is approaching, we would like to kindly ask if our responses sufficiently clarify your concerns or if there are any remaining issues you would like us to address. We appreciate your time and consideration.
> >
> > Best,
> >
> > Authors

---

> > > ### Author Response · Authors · 2024-08-12
> > >
> > > Dear Reviewer R1mD,
> > >
> > > We would like to thank you again for your valuable feedback on our paper.
> > >
> > > As the period for the Author-Reviewer discussion is closing very soon, we would like to use this opportunity to kindly ask if our responses sufficiently clarify your concerns. We sincerely appreciate your time and consideration.
> > >
> > > Best,
> > >
> > > Authors

---

> > > > ### Author Response · Authors · 2024-08-13
> > > >
> > > > Dear Reviewer R1mD,
> > > >
> > > > We would like to thank you again for your valuable feedback on our paper.
> > > >
> > > > As the period for the Author-Reviewer discussion is closing today, we would like to kindly ask if our responses have sufficiently addressed your concerns. We sincerely appreciate your time and consideration.
> > > >
> > > > Best,
> > > >
> > > > Authors

---

### Author Rebuttal · Authors · 2024-08-05

We sincerely thank all reviewers for their thoughtful comments and positive feedbacks. We appreciate their findings of the strengths for this paper, including:

- **our studied problem** (reducing the computational overhead for video diffusion models) is important (Reviewer R1mD) and well validated (Reviewer FT7s);

- **our approach and idea** are sound and well founded (R1mD), are simple and working (Reviewer Do55), make sense and well-suited for the studied task (Reviewer FT7s), and resolve the issue of generating static videos (Reviewer Q9Lc);

- **our results** are excellent, clear, well-explained, and detailed (Reviewer FT7s), are comparable to original model with several diffusion steps (Reviewer Do55), achieve SOTA SVD quality and sampling time in 1-step (Reviewer FT7s), and significantly improve video generation inference speed (Reviewer Q9Lc);

- **our paper** is very clear, well-written, and easy to read (Reviewer Do55, FT7s) with sufficient related work and the right context (Reviewer FT7s).

In the following, we provide detailed answer to the major concern of each reviewer. Additionally, we attach an **one-page PDF** with more visualization results.

---

### Decision · Program_Chairs · 2024-09-25

**Decision:**

Accept (poster)

**Comment:**

This paper has proposed an approach to perform image-to-video inference in a single step through training a GAN model through distillation. The presented approach has strong practical applications, and the idea is simple but works very well. Although the average rating is less than 5, The AC believes all major concerns are already addressed by the rebuttal. The experimental results are compelling, and the evaluation is comprehensive. The AC also examined some issues that reviewers may have overlooked and believes the raised concerns should have been solved. Thus, the AC recommends the acceptance of the paper. The decision of this paper has been discussed with the SAC.